# Research on Improving Satellite Positioning Precision Based on Multi-Frequency Navigation Signals

**DOI:** 10.3390/s22114210

**Published:** 2022-05-31

**Authors:** Ting Kong, Lihua Ma, Guoxiang Ai

**Affiliations:** 1National Astronomical Observatories, Chinese Academy of Sciences, Beijing 100101, China; kongting@nao.cas.cn (T.K.); aigx@nao.cas.cn (G.A.); 2University of Chinese Academy of Sciences, Beijing 100049, China

**Keywords:** multi-frequency navigation, satellite positioning, positioning precision, PDOP

## Abstract

In satellite positioning systems, optimizing navigation satellite constellation and reducing the observation residuals are usually adopted to improve positioning precision and accuracy of the receiver. This paper presents a method to improve positioning precision by using multi-frequency navigation signals. The observation data of CAPS and GPS system are used to simulate the experiment. When the number of downlink frequencies is different, the root mean square of positioning error, improvement percentage, and standard deviation are calculated, respectively. When the number of descending frequencies is k, the root mean square of positioning error in three-dimensional space is 1/$\sqrt{k}$ of that in single frequency. The theoretical derivation and experiment show that the precision of satellite positioning can be effectively improved by using multi-frequency navigation signals. The research work can provide theoretical support and data reference for the future research of satellite positioning.

## 1. Introduction

In satellite positioning systems, positioning precision of the receiver is mainly affected by spatial layout of navigation satellites constellation and measurement errors [1]. Measurement errors generally include satellite clock error, ephemeris error, and errors caused by ionospheric delay, troposphere delay, multipath effect, and receiver thermal noise. The methods such as wide-area difference technology, local-area difference technology, navigation message enhancement, high-precision orbit/clock correction, atmospheric delay correction, and measurement noise suppression are generally adopted to eliminate or weaken the pseudo-range observation error and improve the positioning precision of the receiver. In papers [2,3,4], they introduced the methods of wide-area difference and local-area difference and client performance evaluation. In papers [5,6,7], the observation residual caused by atmospheric delay was improved by improving modeling accuracy. In paper [8], a particle filter tracking algorithm was proposed to suppress multi-path errors under strong and weak signals. The spatial layout of satellite constellation has an important influence on the system performance and positioning precision, and dilution of precision (DOP) is the main parameter to measure the geometric relationship between satellite and receiver. In paper [9,10], the relationship between satellite positioning error and optimal constellation layout was analyzed. By adding satellites and improving the constellation structure, the DOP value and positioning precision can be improved. In the paper [11], a satellite selection method based on NSGA-II genetic algorithm is proposed, which takes into account the Geometric DOP (GDOP) value and the number of selected satellites. In paper [12], a GNSS multi-constellation selection method and an improved GPS real-time satellite selection algorithm were proposed. In paper [13], a constellation selection scheme for the regional satellite positioning system was proposed. In paper [14], it analyzed the influence of multi-constellation on the performance of the navigation system. In paper [15], it demonstrated that the Position DOP (PDOP) value of the GNSS system and the positioning precision can be improved by using LEO satellites. In paper [16], it introduced a GNSS analysis software based on MATLAB software for evaluating positioning results, such as positioning error, number of satellites, DOP value, etc. In paper [17], the weighted GDOP (WDOP) was proposed as an indicator for selecting stars and evaluating positioning precision.

Chinese Area Positioning System (CAPS) is a satellite positioning system based on communication satellites. Satellite navigation signals are broadcast from ground stations and forwarded by communication satellites to reach ground receivers to achieve positioning and navigation [18]. The principle of implementation of the system is shown in Figure 1. The communication satellites in the system include geostationary orbit (GEO) satellites and small inclined geostationary orbit (SIGSO) satellites. During the CAPS system test from 2019 to 2021, the ASIASAT-3S, CHINASAT-10, CHINASAT-12, and APSTAR-6C communication satellites were selected as navigation satellites. Each satellite has more than 20 communication transponders used to transmit CAPS navigation signals. The developed CAPS receiver has 48 signal channels and can simultaneously receive 48 downlink navigation signals. When the constellation layout is not good, three communication satellites can be used to achieve positioning with the aid of elevation [19,20]; when the constellation layout is good, the conventional least squares method can be used to achieve positioning. Combining the forwarding characteristics of the CAPS system and the resource advantages of communication satellites, Li et al. proposed the virtual clock method to effectively eliminate or greatly weaken the ephemeris error of the CAPS system [21]. Ai et al. proposed a method called physical augmentation factor of precision (PAFP) to equivalently improve the spatial layout [22].

The communication signal covers a wide range and has a wide signal frequency band. With the development of communication and positioning technology and as the variety of user needs continues to increase, the combination of communication and navigation has also become a hot research topic. This paper combines the characteristics of the CAPS system and the existing technology to carry out research on using multi-frequency navigation signals to improve positioning precision. Without increasing the number of satellites, by increasing the number of satellite downlink navigation signals, the pseudo-range equations of multi-frequency navigation signals are established to improve the positioning precision. The simulation experiments were carried out using the CAPS measured data and the observation data of the International GNSS Service (IGS) station in Fangshan, Beijing. According to the experimental results, the PDOP value and positioning precision are statistically analyzed to evaluate their improvement. This research provides theoretical support and data reference for the research on the positioning system combining communication satellites and navigation and has certain research significance and application value.

## 2. Methodology and Equations

Under the assumption that the measurement errors are independent and have equal variance, B. Parkinson et al. proposed that the optimal layout of six satellites is that four satellites are equally spaced on the horizontal plane, and the other two are on the zenith; at this time, the minimum value of the geometric dilution of precision (GDOP) is $\sqrt{2}$, as shown in Figure 2 [23]. It is speculated from this that when the optimal layout of N satellites is known, and two satellites are placed in the same position where there was one satellite before, the optimal layout of 2N satellites can be obtained and, at this time, the GDOP is $1/\sqrt{2}$ times the previous.

The CAPS system adopts the virtual clock technology with the pseudo-range differential effect proposed by Li [21], which can eliminate the ephemeris error, clock error, and radial error caused by ionospheric delay and tropospheric delay in measurement errors. Only random errors are left, caused by multi-path effects or receiver thermal noise, etc., which makes the measurement errors of multi-frequency signals independent. It is technically difficult to arrange two satellites at the same position, but it is easy to achieve multi-frequency navigation signals from the same satellite, as shown in Figure 3. The physical augmentation factor of precision (PAFP) proposed by Ai et al. is to improve the spatial layout equivalently through multi-frequency navigation signals [22]. At this time, the PDOP value of the constellation composed of satellites transmitting multi-frequency navigation signals is significantly lower than that of the original system.

### 2.1. Virtual Clock Technology

The virtual clock has a pseudo-range differential effect; in the implementation of virtual clock technology, the integrated baseband of the ground station of the CAPS system is synchronized with the atomic clock of CAPS; the navigation signals are sent from the integrated baseband of the ground station and transmitted to the satellites through the radio frequency transmission channel [21]. The satellite transponder forwards the signals and broadcasts them to the CAPS coverage area, and the user receiver receives the navigation signals to achieve positioning, and the ground station itself also receives the navigation signals at the same time, as shown in Figure 4.

The observed pseudo-range of the user receiver is:(1)$$\rho =c\cdot \left(t_{c}-t_{v}\right)$$
where c is the speed of light; $t_{c}$ is the total delay of the signal sent from the integrated baseband transmitting terminal of the ground station, through the radio frequency transmission channel, space uplink, satellite forwarding, and space downlink to subscriber receiver; $t_{v}$ is the time correction of the virtual clock.

The time correction of the virtual clock is obtained through the virtual clock model, as follows [24]:(2)$$t_{v}=a_{4}{\left(t-t_{0}\right)}^{4}+a_{3}{\left(t-t_{0}\right)}^{3}+a_{2}{\left(t-t_{0}\right)}^{2}+a_{1}\left(t-t_{0}\right)+a_{0}$$
where t is the time when the subscriber receiver receives the signals; $t_{0}$ is the start time of the virtual clock model; $a_{0}$, $a_{1}$, $a_{2}$, $a_{3}$, and $a_{4}$ are detected by the ground monitoring station.

The ground station forwards the model parameters to the subscriber receiver through the navigation message, and the CAPS navigation information is generated by the ground station, including time information, synchronization information, satellite orbit parameters, virtual clock model parameters, wide area differential information, local pseud-range differential information, local position difference data, etc.

Regarding the virtual clock deviation caused by the satellite position deviation, Li et al. also made theoretical analysis and actual measurement [21]. When there is 1 km difference between the actual position of the satellite and the predicted position of the satellite by ephemeris data, the receiver uses virtual clock technology to correct the ephemeris error, and the result of the positioning solution is almost the same as the result calculated by using the actual position of the satellite, which shows that the virtual clock technology can cancel or greatly weaken the ephemeris error.

### 2.2. Physical Augmentation Factor of Precision (PAFP)

Combining the resource advantages of the communication satellites of the CAPS system, Ai et al. proposed a method of using multiple frequencies to broadcast navigation signals separately, which can equivalently improve the spatial layout of the constellation [22]. The CAPS system selects communication band (including C-band and Ku-band) to transmit satellite navigation signals. There are more than 20 communication transponders on numerous communication satellite to transmit CAPS navigation signals. The working frequency bands and available frequency points are shown in Table 1.

Assuming that the number of visible satellites is n, each satellite sends down one navigation signal, and the coefficient matrix obtained by linearizing the pseudo-range observation equation is:(3)$$H=\left[\begin{array}{ccc} \begin{array}{c} {\mathrm{hx}}_{1}\\ {\mathrm{hx}}_{2}\\ {\mathrm{hx}}_{3} \end{array} & \begin{array}{c} {\mathrm{hy}}_{1}\\ {\mathrm{hy}}_{2}\\ {\mathrm{hy}}_{3} \end{array} & \begin{array}{c} {\mathrm{hz}}_{1}\\ {\mathrm{hz}}_{2}\\ {\mathrm{hz}}_{3} \end{array}\\ \vdots & \vdots & \vdots \\ {\mathrm{hx}}_{i} & {\mathrm{hy}}_{i} & {\mathrm{hz}}_{i} \end{array}\begin{array}{c} \begin{array}{c} 1\\ 1\\ 1 \end{array}\\ \vdots \\ 1 \end{array}\right]$$
where $\left[{\mathrm{hx}}_{i},{\mathrm{hy}}_{i},{\mathrm{hz}}_{i}\right]$ (i = 1, 2, 3, …, n) is the cosine vector from the receiver position to each satellite, and n is the number of satellites involved in the calculation. Then, the weight coefficient matrix is:(4)$$D={\left(H^{T}H\right)}^{-1}={\left[\begin{array}{cccc} \sum_{i=1}^{n}{\mathrm{hx}}_{i}{}^{2} & \sum_{i=1}^{n}{\mathrm{hx}}_{i}{\mathrm{hy}}_{i} & \sum_{i=1}^{n}{\mathrm{hx}}_{i}{\mathrm{hz}}_{i} & \sum_{i=1}^{n}{\mathrm{hx}}_{i}\\ \sum_{i=1}^{n}{\mathrm{hx}}_{i}{\mathrm{hy}}_{i} & \sum_{i=1}^{n}{\mathrm{hy}}_{i}{}^{2} & \sum_{i=1}^{n}{\mathrm{hy}}_{i}{\mathrm{hz}}_{i} & \sum_{i=1}^{n}{\mathrm{hy}}_{i}\\ \sum_{i=1}^{n}{\mathrm{hx}}_{i}{\mathrm{hz}}_{i} & \sum_{i=1}^{n}{\mathrm{hy}}_{i}{\mathrm{hz}}_{i} & \sum_{i=1}^{n}{\mathrm{hz}}_{i}{}^{2} & \sum_{i=1}^{n}{\mathrm{hz}}_{i}\\ \sum_{i=1}^{n}{\mathrm{hx}}_{i} & \sum_{i=1}^{n}{\mathrm{hy}}_{i} & \sum_{i=1}^{n}{\mathrm{hz}}_{i} & n \end{array}\right]}^{-1}=\left[\begin{array}{ccc} P_{11} & P_{12} & \begin{array}{cc} P_{13} & P_{14} \end{array}\\ P_{21} & P_{22} & \begin{array}{cc} P_{23} & P_{24} \end{array}\\ \begin{array}{c} P_{31}\\ P_{41} \end{array} & \begin{array}{c} P_{32}\\ P_{42} \end{array} & \begin{array}{cc} \begin{array}{c} P_{33}\\ P_{43} \end{array} & \begin{array}{c} P_{34}\\ P_{44} \end{array} \end{array} \end{array}\right]$$

Then, the PDOP value can be expressed as:(5)$$\mathrm{PDOP}=\sqrt{P_{11}+P_{22}+P_{33}}$$
where $P_{\mathrm{ii}}$ (i = 1, 2, 3, 4) is the diagonal element of the weight coefficient matrix of the pseudo-range observation equation.

In a satellite positioning system, the variance of the pseudo-range observation error is amplified by the weight coefficient matrix to obtain the variance of the positioning error [1]. The relationship can be expressed as:(6)$$\left[\begin{array}{ccc} \sigma_{x}{}^{2} & & \\ & \sigma_{y}{}^{2} & \\ & & \begin{array}{cc} \sigma_{z}{}^{2} & \\ & \sigma_{t}{}^{2} \end{array} \end{array}\right]=\left[\begin{array}{ccc} P_{11} & & \\ & P_{22} & \\ & & \begin{array}{cc} P_{33} & \\ & P_{44} \end{array} \end{array}\right]{*\sigma }_{e}{}^{2}$$
where $\sigma_{x}{}^{2}$, $\sigma_{y}{}^{2}$, $\sigma_{z}{}^{2}$, and $\sigma_{t}{}^{2}$ are the variance of the positioning errors in the X-axis, Y-axis, Z-axis directions, and clock difference in the ECEF coordinate system, respectively; $P_{\mathrm{ii}}$ (i = 1, 2, 3, 4) is the diagonal element of the weight matrix of the pseudo-range observation equation.

Then, the standard deviation of the positioning error in three-dimensional space is:(7)$$\sigma =\sqrt{\sigma_{x}{}^{2}+\sigma_{y}{}^{2}+\sigma_{z}{}^{2}}=\sqrt{P_{11}+P_{22}+P_{33}}{*\sigma }_{e}$$

When the number of visible satellites is n, each visible satellite sends down multi-frequency navigation signals, the number of downlink frequency is k, and its coefficient matrix is:(8)$$H_{B}=\left[\begin{array}{cccc} {\mathrm{hx}}_{11} & {\mathrm{hy}}_{11} & {\mathrm{hz}}_{11} & 1\\ {\mathrm{hx}}_{21} & {\mathrm{hy}}_{21} & {\mathrm{hz}}_{21} & 1\\ \vdots & \vdots & \vdots & \vdots \\ {\mathrm{hx}}_{i1} & {\mathrm{hy}}_{i1} & {\mathrm{hz}}_{i1} & 1\\ {\mathrm{hx}}_{12} & {\mathrm{hy}}_{12} & {\mathrm{hz}}_{12} & 1\\ {\mathrm{hx}}_{22} & {\mathrm{hy}}_{22} & {\mathrm{hz}}_{22} & 1\\ \vdots & \vdots & \vdots & \vdots \\ {\mathrm{hx}}_{i2} & {\mathrm{hy}}_{i2} & {\mathrm{hz}}_{i2} & 1\\ \vdots & \vdots & \vdots & \vdots \\ {\mathrm{hx}}_{ij} & {\mathrm{hy}}_{ij} & {\mathrm{hz}}_{ij} & 1 \end{array}\right]$$
where $\left[{\mathrm{hx}}_{\mathrm{ij}},{\mathrm{hy}}_{\mathrm{ij}},{\mathrm{hz}}_{\mathrm{ij}}\right]$ (i = 1, 2, …, n; j = 1, 2, …, k) is the cosine vector of the direction from the receiver to the i-th satellite that transmits the j-th signal, so:(9)$${\mathrm{hx}}_{i1}={\mathrm{hx}}_{i2}=\ldots ={\mathrm{hx}}_{\mathrm{ij}}$$
(10)$${\mathrm{hy}}_{i1}={\mathrm{hy}}_{i2}=\ldots ={\mathrm{hy}}_{\mathrm{ij}}$$
(11)$${\mathrm{hz}}_{i1}={\mathrm{hz}}_{\mathrm{ij}}=\ldots ={\mathrm{hz}}_{\mathrm{ij}}$$

According to Equations (6)–(9), the weight coefficient matrix of the pseudo-range observation equation can be obtained as:(12)$$D_{B}={\left(H_{B}^{T}H_{B}\right)}^{-1}={\left[\begin{array}{cccc} k\sum_{i=1}^{n}{\mathrm{hx}}_{i1}{}^{2} & k\sum_{i=1}^{n}{\mathrm{hx}}_{i1}{\mathrm{hy}}_{i1} & k\sum_{i=1}^{n}{\mathrm{hx}}_{i1}{\mathrm{hz}}_{i1} & k\sum_{i=1}^{n}{\mathrm{hx}}_{i1}\\ k\sum_{i=1}^{n}{\mathrm{hx}}_{i1}{\mathrm{hy}}_{i1} & k\sum_{i=1}^{n}{\mathrm{hy}}_{i1}{}^{2} & k\sum_{i=1}^{n}{\mathrm{hy}}_{i1}{\mathrm{hz}}_{i1} & k\sum_{i=1}^{n}{\mathrm{hy}}_{i1}\\ k\sum_{i=1}^{n}{\mathrm{hx}}_{i1}{\mathrm{hz}}_{i1} & k\sum_{i=1}^{n}{\mathrm{hy}}_{i1}{\mathrm{hz}}_{i1} & k\sum_{i=1}^{n}{\mathrm{hz}}_{i1}{}^{2} & k\sum_{i=1}^{n}{\mathrm{hz}}_{i1}\\ k\sum_{i=1}^{n}{\mathrm{hx}}_{i1} & k\sum_{i=1}^{n}{\mathrm{hy}}_{i1} & k\sum_{i=1}^{n}{\mathrm{hz}}_{i1} & k\cdot n \end{array}\right]}^{-1}=\frac{1}{k}\left[\begin{array}{ccc} P_{11} & P_{12} & \begin{array}{cc} P_{13} & P_{14} \end{array}\\ P_{21} & P_{22} & \begin{array}{cc} P_{23} & P_{24} \end{array}\\ \begin{array}{c} P_{31}\\ P_{41} \end{array} & \begin{array}{c} P_{32}\\ P_{42} \end{array} & \begin{array}{cc} \begin{array}{c} P_{33}\\ P_{43} \end{array} & \begin{array}{c} P_{34}\\ P_{44} \end{array} \end{array} \end{array}\right]$$

Then, its PDOP value can be expressed as:(13)$${\mathrm{PDOP}}_{B}=\sqrt{\frac{P_{11}+P_{22}+P_{33}}{k}=}\frac{\mathrm{PDOP}}{\sqrt{k}}$$

It is seen that the ratio of the improved PDOP value to the original PDOP value is $\frac{1}{\sqrt{k}}$. When the original PDOP value is constant, the improved PDOP value is inversely proportional to the square root of the number of downlink frequencies. This ratio is called the physical augmentation factor of precision (PFAP).

Then, when each visible satellite sends down multiple-frequency navigation signals, the standard deviation of the positioning error in three-dimensional space is:(14)$$\sigma_{B}=\sqrt{\sigma_{x}{}^{2}+\sigma_{y}{}^{2}+\sigma_{z}{}^{2}}=\frac{\sqrt{P_{11}+P_{22}+P_{33}}}{k}\times \sigma_{e\_B}$$

### 2.3. Positioning Solution of Multi-Frequency Receiver

The baseband system of the C/Ku dual-band navigation receiver in the user segment in the CAPS has 48 signal channels, which can simultaneously receive 48 downlink navigation signals, as shown in Figure 5. It uses virtual clock technology to correct satellite orbit errors, clock errors, and errors caused by atmospheric delay in the measurement pseudo-range and solves the receiver position by establishing pseudo-range equations to solve satellite positioning.

The process of satellite positioning using multi-frequency pseudo-range observations is as follows.

The pseudo-range observation equation of the navigation satellite can be expressed as:(15)$$\rho_{\mathrm{ij}}=c\cdot \left(t_{c}-t_{v}\right)=\sqrt{{\left(X_{i}-X\right)}^{2}+{\left(Y_{i}-Y\right)}^{2}+{\left(Z_{i}-Z\right)}^{2}}+\varepsilon_{\mathrm{ij}}$$
where $\rho_{\mathrm{ij}}$ is the distance from the receiver to the i-th satellite that transmits the j-th signal, that is, the pseudo-range from the satellite to the receiver; c is the speed of light; $t_{c}$ is the total delay of the signal sent from the integrated baseband transmitting terminal of the ground station, through the radio frequency transmission channel, space uplink, satellite forwarding, and space downlink to the integrated baseband accept terminal of the subscriber receiver; $t_{v}$ is the time correction of the virtual clock; ($X_{i}$, $Y_{i}$, $Z_{i}$) is the position coordinate of the i-th satellite; (X, Y, Z) is the receiver position coordinates; and $\varepsilon_{\mathrm{ij}}$ is gaussian-distributed random errors due to multipath effects and receiver thermal noise, etc.

The pseudo-range observation equation is expanded by the first-order Taylor series at the initial point O $\left(X_{0},Y_{0},Z_{0}\right)$. After omitting the higher-order terms, we can get:(16)$$\rho_{\mathrm{ij}}-\rho_{i}\left(0\right)=H\cdot \mathrm{dX}+\varepsilon_{\mathrm{ij}}$$

Its coefficient matrix is:(17)$$H=\left[\begin{array}{cccc} {\mathrm{hx}}_{11} & {\mathrm{hy}}_{11} & {\mathrm{hz}}_{11} & 1\\ {\mathrm{hx}}_{21} & {\mathrm{hy}}_{21} & {\mathrm{hz}}_{21} & 1\\ \vdots & \vdots & \vdots & \vdots \\ {\mathrm{hx}}_{i1} & {\mathrm{hy}}_{i1} & {\mathrm{hz}}_{i1} & 1\\ {\mathrm{hx}}_{12} & {\mathrm{hy}}_{12} & {\mathrm{hz}}_{12} & 1\\ {\mathrm{hx}}_{22} & {\mathrm{hy}}_{22} & {\mathrm{hz}}_{22} & 1\\ \vdots & \vdots & \vdots & \vdots \\ {\mathrm{hx}}_{i2} & {\mathrm{hy}}_{i2} & {\mathrm{hz}}_{i2} & 1\\ \vdots & \vdots & \vdots & \vdots \\ {\mathrm{hx}}_{ij} & {\mathrm{hy}}_{ij} & {\mathrm{hz}}_{ij} & 1 \end{array}\right]$$
where $\left[{\mathrm{hx}}_{\mathrm{ij}},{\mathrm{hy}}_{\mathrm{ij}},{\mathrm{hz}}_{\mathrm{ij}}\right]$ (i = 1, 2, …, n; j = 1, 2, …, k) is the cosine vector of the direction from the receiver to the i-th satellite that transmits the j-th signal.

The difference between the observed pseudo-range and the predicted pseudo-range value at the initial point can be expressed as:(18)$$B=\left[\begin{array}{c} \begin{array}{c} \rho_{11}-\rho_{1}\left(0\right)\\ \rho_{21}-\rho_{2}\left(0\right)\\ \vdots \end{array}\\ \begin{array}{c} \rho_{i1}-\rho_{i}\left(0\right)\\ \vdots \\ \rho_{12}-\rho_{1}\left(0\right) \end{array}\\ \begin{array}{c} \rho_{22}-\rho_{2}\left(0\right)\\ \vdots \\ \begin{array}{c} \rho_{i2}-\rho_{i}\left(0\right)\\ \vdots \\ \begin{array}{c} \rho_{\mathrm{ij}}-\rho_{i}\left(0\right) \end{array} \end{array} \end{array} \end{array}\right]$$
where $\rho_{\mathrm{ij}}$ is the pseudo-range observation value; $\rho_{i}\left(0\right)$ is the pseudo-range value predicted by the initial point, which can be expressed as:(19)$$\rho_{i}\left(O\right)=\sqrt{{\left(X_{0}-X_{i}\right)}^{2}+{\left(Y_{0}-Y_{i}\right)}^{2}+{\left(Z_{0}-Z_{i}\right)}^{2}}$$
where $\left(X_{0},Y_{0},Z_{0}\right)$ are the coordinates of the initial point, and ($X_{i}$, $Y_{i}$, $Z_{i}$) are the position coordinates of the i-th satellite.

The estimated value of the offset calculated by the least square algorithm is:(20)$$\mathrm{dX}=\left[\begin{array}{c} \Delta X\\ \Delta Y\\ \begin{array}{c} \Delta Z\\ \Delta \delta t \end{array} \end{array}\right]={\left(H^{T}H\right)}^{-1}H^{T}B$$

Add the current offset to the initial value to get the initial value of the next iteration ($X_{0}$ + ΔX, $Y_{0}$ + ΔY, $Z_{0}$ + ΔZ). After reaching the threshold, stop the iteration and get the current receiver position estimate.

## 3. Experimental Data and Analysis Indicators

### 3.1. Observation Data of CAPS System

The CAPS system was in the test period in April 2021. The system uses two GEO satellites and a SIGSO satellite, namely APSTAR-7 (76.5° E), CHINASAT-10 (110.5° E), and ASIASAT-3S (147.5° E), to form a navigation constellation. The measured data of the CAPS system from 17:00 to 18:00 on 1 April 2021, with a sampling interval of one second, were collected at the Wuqing station in Tianjin, China. The measured pseudo-ranges in this data are corrected by virtual clock technology, which eliminates or greatly reduces correlated errors, including ephemeris errors, clock errors, errors caused by atmospheric delays, etc. We simulate the pseudo-range of multi-frequency measurement by adding random noise that satisfies the Gaussian distribution on the basis of the single-frequency measured pseudo-range. The position coordinates of the observation station are obtained by using the long-term static relative positioning method of commercial RTK.

### 3.2. Observational Data from GNSS Systems

To verify the degree of improvement of the positioning precision of this method on the GNSS system, the GPS satellite ephemeris data provided by the IGS station on 8 September 2021 at the Fangshan in Beijing, China for 24 h with a sampling interval of 30 s was selected as the satellite constellation of the simulation experiment. The visible number of satellites during the observation time is shown in Figure 6.

At present, the GNSS system can correct the ephemeris error and clock error by using the precise ephemeris and precise clock error products provided by IGS, and use the dual-frequency ionospheric elimination combined model to weaken the ionospheric error; use the Saastamonien model to correct the tropospheric delay. In the simulation experiment of GNSS, on the premise that the correlation models are used to weaken the correlation system error, a random error with a variance of 1 and a mean of 0 that satisfies the Gaussian distribution is added to the geometric distance from the position coordinates of the satellite to the position coordinates of the Fangshan station to simulate the measurement pseudo-ranges of multiple frequency points. The position coordinates of the observation station are the coordinates of the IGS annual solution.

### 3.3. Analysis Indicators

#### 3.3.1. Root Mean Square Error (RMSE)

The root mean square error is used to measure the positioning precision of the positioning system in the three directions of east, north, zenith, and three-dimensional space. That is, the difference between the calculated estimated value and the reference value is calculated as:(21)$$\mathrm{RMSE}=\sqrt{\frac{\sum_{i=1}^{n}{\left(x_{i}-x_{0}\right)}^{2}}{n}}$$
(22)$$3\mathrm{DRMSE}=\sqrt{\frac{\sum_{i=1}^{n}{\left(x_{i}-x_{0}\right)}^{2}+{\left(y_{i}-y_{0}\right)}^{2}+{\left(z_{i}-z_{0}\right)}^{2}}{n}}$$
where i is the epoch time; n is the total epoch; $x_{i},{\;y}_{i},{\;z}_{i}$ are the estimated value scalculated in the three directions of the north, east, and zenith, respectively, at the i-th epoch time; and $x_{0},{\;y}_{0},{\;z}_{0}$ are the reference values of the observation station.

#### 3.3.2. Improvement Percentage (IP)

The improvement percentage is used to reflect the improvement of the positioning precision; that is, when the number of downlink frequencies is different, the calculated positioning precision is improved relative to the single frequency. The calculation formula is:(23)$$\mathrm{IP}=\frac{r_{i}-r_{0}}{r_{0}}\times 100\%$$
where $r_{i}$ is the positioning precision when the frequency number is i, i = 1, 2, …, n, n is the frequency number; and $r_{0}$ is the positioning precision when the frequency is single.

#### 3.3.3. Standard Deviation (STD)

The standard deviation is used to reflect the degree of dispersion of the positioning results, that is, the difference between the calculated estimated value and the mean of the estimated value. The calculation formula is:(24)$$\mathrm{STD}=\sqrt{\frac{\sum_{i=1}^{n}{\left(x_{i}-\bar{x}\right)}^{2}}{n}}$$
where i is the epoch time; n is the total epoch; $x_{i}$ is the estimated value calculated at the i-th epoch time; and $\bar{x}$ is the average of the estimates computed at all epochs.

## 4. Results

The navigation receiver of the CAPS system has 48 signal channels and can receive at least 12 channels of C-band navigation signals at the same time. Assuming that the number of downlink frequencies of each visible satellite is at most 12, when the downlink frequencies are different, the maximum, minimum, and average values of PDOP in the CAPS system and GNSS system are counted respectively. The results are shown in Table 2.

To more intuitively represent the relationship between the PDOP value and the number of downlink frequencies, the changes of the PDOP value of CAPS and GNSS with the number of frequencies are plotted in Figure 7.

As shown in Table 2 and Figure 7, the use of multi-frequency navigation signals for satellite positioning can significantly reduce the PDOP value of the satellite navigation system. When the number of downlink frequencies is k, the PDOP value is about $1/\sqrt{k}$ of the original PDOP value. In the CAPS system, when the number of downlink frequency of each satellite is 1, that is, when multi-frequency is not used, the maximum value of the PDOP value is 6.85, the minimum value is 6.82, and the average value is 6.83; when the downlink frequencies of each satellite is 12, the maximum PDOP value drops to 1.98, the minimum value drops to 1.97, and the average value drops to 1.97. In the GNSS system, when the number of the downlink frequency of each satellite is 1, the maximum PDOP value is 2.91, the minimum value is 1.17, and the average value is 1.76; when the number is 12, its maximum PDOP value drops to 0.84, its minimum value drops to 0.34, and its average value drops to 0.51.

Based on the above satellite constellation space layout, take the observation station as the coordinate origin, and calculate the position coordinates of the receiver in the three directions of the north, east, and zenith at each moment. When the number of downlink frequencies is different, the root means square error, improvement percentage, and standard deviation of the three directions are calculated separately. The statistical results of the CAPS system are shown in Table 3, and the statistical results of the GNSS system are shown in Table 4; the variation of statistical results with the number of downlink frequencies are shown in Figure 8.

As shown in Table 3 and Table 4 and Figure 8, when the number of downlink frequencies is k, the positioning precision of the system is about 1/$\sqrt{k}$ of the original. In the CAPS system, the elevation-assisted method is used for the positioning calculation, and the positioning error in the height is less than 0.01 m, so it is not counted here. When the number of downlink frequencies is 12, in the CAPS system, the positioning precision in the east and north directions are 0.57 and 2.18, respectively, which are 52% and 68% improved compared with the single frequency; the positioning precision of the two-dimensional space is improved by 67% compared with the single frequency. In the GNSS system, the positioning precision in the east, north, and zenith directions are 0.17, 0.23, 0.43, respectively, which are improved by 72%, 71%, and 71% compared with the single frequency; the positioning precision of the three-dimensional space is improved by 71% compared with the single frequency. The system standard deviation decreases as the number of frequencies increases, and it is seen that the accuracy and stability of the calculation results are improved.

## 5. Discussion

This paper introduces a method of using multi-frequency navigation signals to improve positioning precision and uses the observation data of the CAPS system and GNSS system to conduct simulation experiments. Statistical analysis was carried out on the experimental results in terms of root mean square error, improvement percentage, and standard deviation. It is verified that the method can improve the positioning precision, improve the stability of the positioning results, and can effectively reduce the PDOP value. The method of improving the positioning precision in this paper is different from the study of multi-scale ionospheric irregularity proposed in the paper [5]; it is also different from the improvement of ionospheric model in the paper [6]; it is also different from the improved solution of the whole week ambiguity proposed in paper [7]. Instead, the multi-channel navigation signal is received by the multi-frequency receiver, the distance from the satellite to the receiver is measured, and multiple pseudo-distance equations are established to achieve the purpose of positioning precision. This paper analyzes the improvement degree of multi-frequency on positioning precision from the time domain, and quantitatively analyzes the positioning results according to the number of downlink frequencies, which is different from the analysis of the influence of multi-frequency on the positioning results from the frequency domain in [25]. It is different from the one proposed by paper [8] to optimize satellite constellation from the angle of altitude angle to achieve the purpose of improving the PDOP value. It is also different from the weighted PDOP value method proposed in the paper [26] as the basis for selecting the best constellation. By increasing the number of downlink navigation signals of a single satellite without increasing the number of satellites, the method effectively improves the spatial layout of satellites and reduces the PDOP value. Through theoretical research and simulation analysis, it is found that when each satellite sends down multiple navigation signals, the ratio of the improved PDOP to the original PDOP is inversely proportional to the square root of the number of downlink navigation signals.

## 6. Conclusions

Based on the existing equipment and technology of the CAPS system, this paper introduces a method to improve the positioning precision through multi-frequency navigation signals, which can effectively reduce the PDOP value and improve the positioning precision. In the simulation results of this paper, in the CAPS system, when the number of downlink multi-frequency navigation signals from each satellite is 12, compared with single-frequency navigation signals, the average PDOP value drops from 6.83 to 1.97, the positioning precision in two-dimensional space is improved by 67%. Because the test data is collected in the CAPS test phase, the positioning error is large and does not represent the final positioning precision of the system. In the GNSS system, when the number of downlink frequencies is 12, and when the daily average PDOP of the satellite constellation is basically kept at about 2, the PDOP value can be reduced to less than 1 through multi-frequency navigation signals, and the positioning precision in 3D space is improved by 71%; the standard deviation decreases with the increase of the number of frequencies, and the improvement effect of positioning precision and stability is obvious. This research provides theoretical support and data reference for future satellite positioning technology combining communication satellites and navigation and has theoretical research and application value.

## Figures and Tables

**Figure 1 sensors-22-04210-f001:**
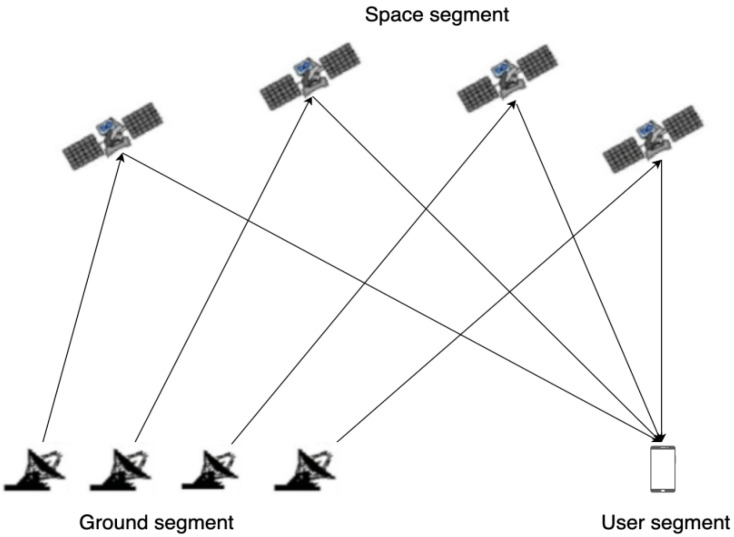
Schematic diagram of the principle of the CAPS system.

**Figure 2 sensors-22-04210-f002:**
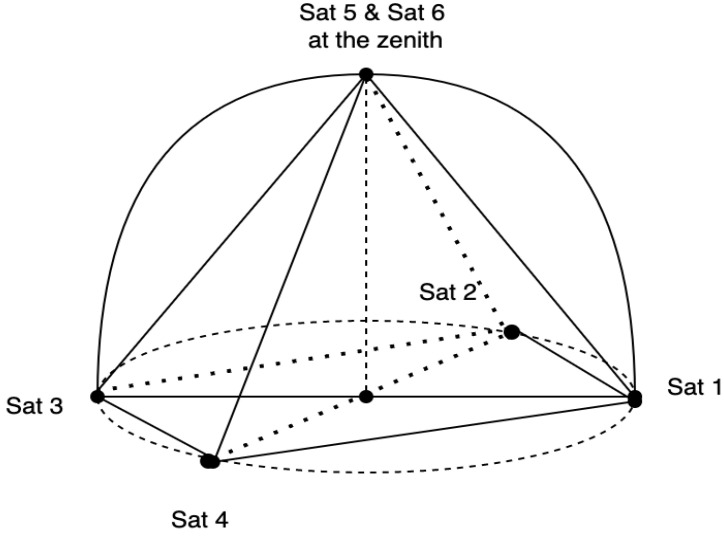
Layout of the six satellites with the smallest GDOP value.

**Figure 3 sensors-22-04210-f003:**
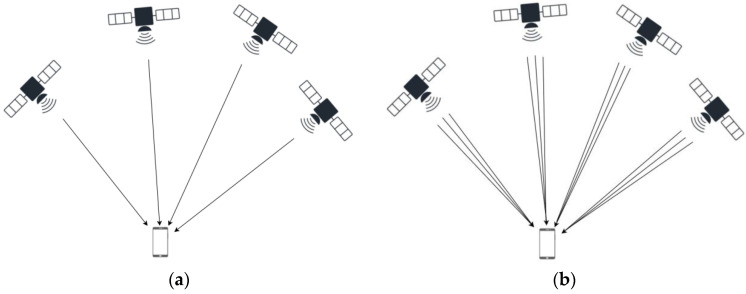
Comparison of downlink single-frequency and multi-frequency signals from in-orbit satellites. (**a**) Each satellite sends down one navigation signal; (**b**) each satellite sends down multiple navigation signals with multi-frequencies.

**Figure 4 sensors-22-04210-f004:**
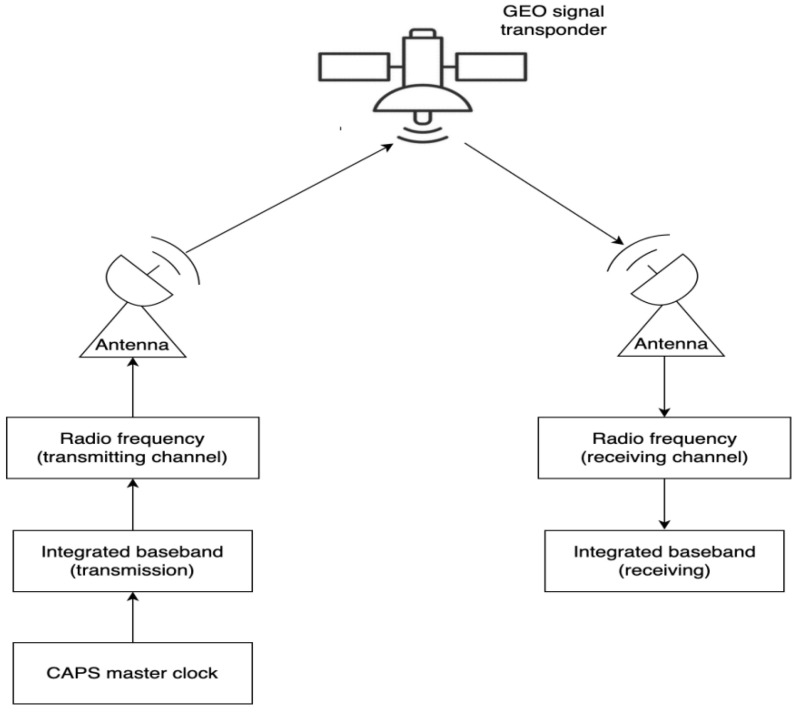
The realization process of the ground station transmitting and receiving the navigation signals.

**Figure 5 sensors-22-04210-f005:**
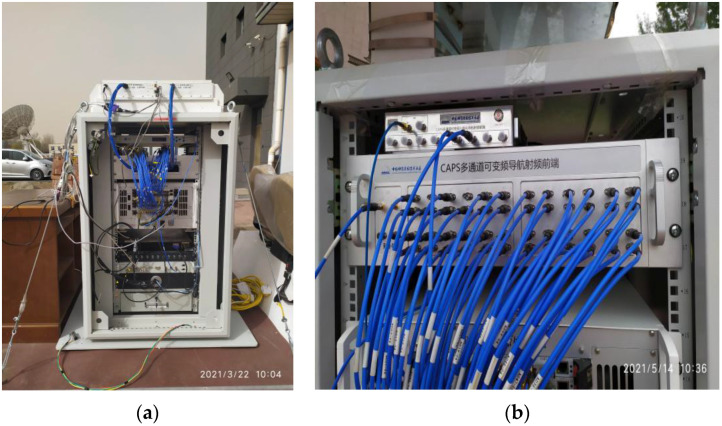
CAPS navigation receiver: (**a**) is the physical diagram of the CAPS receiver, including multi-beam antenna, frequency converter, radio frequency device, signal channel, intelligent control terminal (including memory, CPU, display), power supply, etc.; (**b**) is a variable frequency navigation radio frequency device with 48 channels.

**Figure 6 sensors-22-04210-f006:**
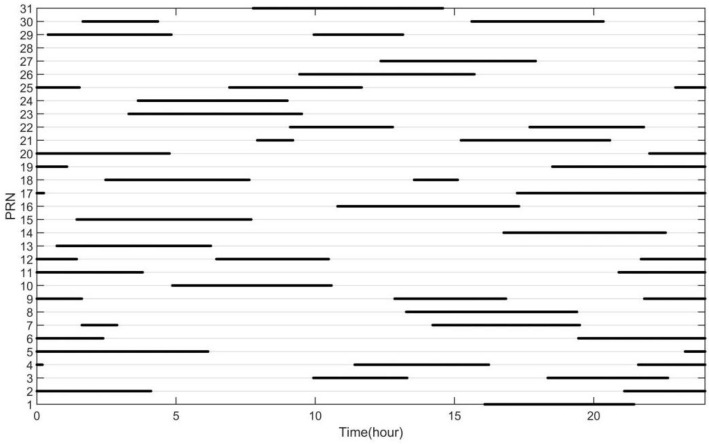
The situation of the visible satellites at the Fangshan station in Beijing on 8 September 2021.

**Figure 7 sensors-22-04210-f007:**
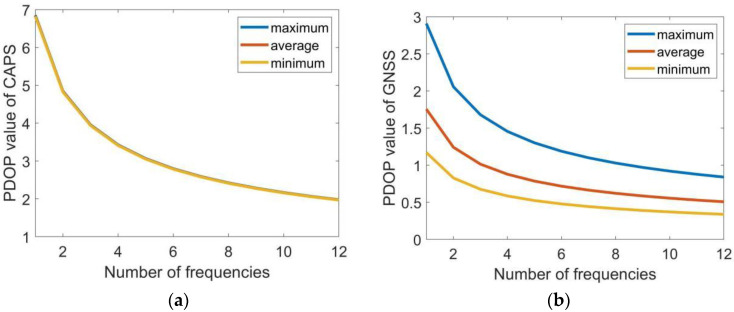
The change of the PDOP value with the increase of the number of frequencies. Subfigures (**a**,**b**) are corresonding to the CAPS system and the GNSS system, respectively.

**Figure 8 sensors-22-04210-f008:**
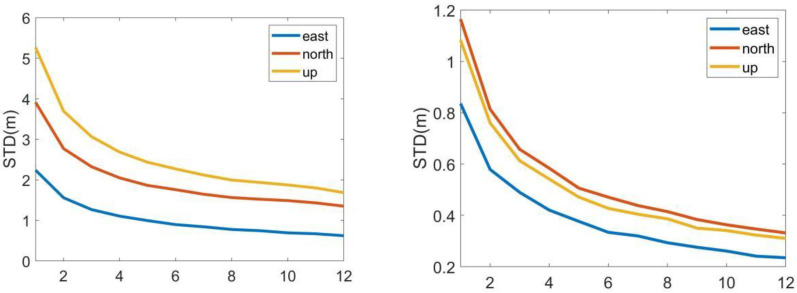
Positioning precision and improvement of multi-frequency navigation signals in CAPS system and GNSS system. Subfigures (**a**) at left column are standard deviation and the root mean square error in three directions and root mean square error of three-dimensional space change with the number of downlink frequencies in CAPS system; abd subfigures (**b**) at right column are standard deviation and the root mean square error in three directions and root mean square error of three-dimensional space change with the number of downlink frequencies in GNSS system.

**Table 1 sensors-22-04210-t001:** Frequency band and the number of available frequency points.

The Name of the Band	Up-Band (MHz)	Down-Band (MHz)	Available Bandwidth (MHz)	The Number of Available Frequency
Standard C-band	5925~6425	3700~4200	500	78
Extended C-band	6425~6725	3400~3700	500	
Standard Ku-band	14,000~14,500	12,250~12,750	500	139
Extended Ku-band	13,750~14,000	10,950~11,200	250	
Planning Ku-Band	12,750~13,250	10,700~10,950, 11,200~11,450	500	
Broadcast Ku band	17,300~17,800	11,700~12,200	500	

**Table 2 sensors-22-04210-t002:** Statistics of PDOP values of multi-frequency navigation signals in CAPS system and GNSS system.

The Number of Downlink Frequencies	CAPS			GNSS		
	Max	Min	Average	Max	Min	Average
1	6.85	6.82	6.83	2.91	1.17	1.76
2	4.84	4.82	4.83	2.06	0.83	1.24
3	3.96	3.94	3.94	1.68	0.68	1.01
4	3.43	3.41	3.41	1.45	0.58	0.88
5	3.06	3.05	3.05	1.3	0.52	0.79
6	2.8	2.78	2.79	1.19	0.48	0.72
7	2.59	2.58	2.58	1.1	0.44	0.66
8	2.42	2.41	2.41	1.03	0.41	0.62
9	2.28	2.27	2.28	0.97	0.39	0.59
10	2.17	2.16	2.16	0.92	0.37	0.56
11	2.07	2.06	2.06	0.88	0.35	0.53
12	1.98	1.97	1.97	0.84	0.34	0.51

**Table 3 sensors-22-04210-t003:** Statistical results of the CAPS system.

The Number of Downlink Frequencies	RMS (m)			IP (%)			STD (m)	
	E	N	2D	E	N	2D	E	N
1	1.2	6.83	6.93	-	-	-	2.24	3.91
2	0.93	4.79	4.88	23	30	30	1.56	2.77
3	0.81	3.97	4.05	33	42	42	1.27	2.32
4	0.75	3.48	3.56	38	49	49	1.1	2.05
5	0.69	3.16	3.23	42	54	53	0.99	1.86
6	0.66	2.94	3.02	45	57	56	0.9	1.76
7	0.64	2.75	2.82	47	60	59	0.84	1.64
8	0.62	2.59	2.66	49	62	62	0.78	1.56
9	0.61	2.51	2.58	49	63	63	0.75	1.52
10	0.6	2.43	2.5	51	64	64	0.69	1.49
11	0.59	2.33	2.41	51	66	65	0.67	1.43
12	0.57	2.18	2.26	52	68	67	0.62	1.35

**Table 4 sensors-22-04210-t004:** Statistical results of the GNSS system.

The Number of Downlink Frequencies	RMS (m)				IP (%)				STD (m)		
	E	N	U	3D	E	N	U	3D	E	N	U
1	0.62	0.8	1.49	1.8	-	-	-	-	0.84	1.16	1.08
2	0.44	0.56	1.04	1.26	29	29	30	30	0.58	0.81	0.76
3	0.36	0.46	0.84	1.03	41	43	43	43	0.49	0.66	0.61
4	0.31	0.4	0.75	0.9	50	50	50	50	0.42	0.58	0.54
5	0.28	0.36	0.65	0.79	55	55	56	56	0.38	0.51	0.47
6	0.25	0.33	0.59	0.72	59	59	60	60	0.33	0.47	0.43
7	0.23	0.3	0.57	0.68	62	62	62	62	0.32	0.44	0.4
8	0.22	0.28	0.53	0.64	65	64	64	64	0.29	0.41	0.39
9	0.21	0.26	0.49	0.59	66	67	67	67	0.28	0.38	0.35
10	0.19	0.25	0.47	0.57	69	68	68	68	0.26	0.36	0.34
11	0.18	0.25	0.44	0.54	71	69	70	70	0.24	0.35	0.32
12	0.17	0.23	0.43	0.52	72	71	71	71	0.24	0.33	0.31

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
