# Peer review of "Research on Improving Satellite Positioning Precision Based on Multi-Frequency Navigation Signals"

_sensors, 2022, doi:10.3390/s22114210_

Round 1

Reviewer 2 Report

 The authors should clarify terminology: from the examples provided my guess is that accuracy (systematic deviation) remains the same, precision (scatter) improves. It is quite obvious that for the GEOs solutions are not accurate (huge trends in Y and Z) but are very precise. In fact is there a need to present the GEO analysis? It weakens the paper.

There is another thing that makes reading the paper very hard. For example when saying the positioning error of the X coordinate is reduced by 0.4 m my guess is that authors are  talking about some sort of standard deviation (std). How is that computed? Is it an average of instantaneous stds, does it result from quadratic deviations to a smoothed time series? Please rewrite the paper to clarify what you are talking about.

----some minor notes

Authors use extensively "the satellite positioning" when I think removing the "the" would be recommended.

"can be seen" is used at least once. It should also be replaced by an affirmative stance: "It is seen" or something similar since there is no ambiguity in what is said.

The section on linearization of least squares could be shortened but then it is only one page so I have no objections to the authors to leave it like that.

Reviewer 3 Report

Line 16: In my opinion, the population counting the samples from 24 hours of observing the positions of the satellites is insufficient to perform the optimization. more population 3-5 days are required with samples from 24 hours of system operation.

If you do not have such observations, please leave it as it is. I'm not going to be picky about this.

Line 186 fig 3. To better illustrate the improvement of the signal quality depending on the number of frequencies used, it is proposed to visualize the results in the frequency domain of the changes occurring before and after the application of the multi-frequency model of the quality improvement of DOP coefficients and position. An example of the method application is presented in:

Felski, A .; Jaskólski, K .; Zwolak, K .; Piskur, P. Analysis of Satellite Compass Error's Spectrum. Sensors 2020, 20, 4067. https://doi.org/10.3390/s20154067

In this case, in the frequency domain, you will record the changes in accuracy and determine the harmonic components of the changes in the accuracy of the coordinates in the x and y and z planes.

Line 226, fig 5: You can also present the coefficient change values ​​in the frequency domain using the Fast Fourier Transform. In this way, you will present the frequency of occurrence of the individual DOP coefficients.

Line 239, fig 6: The coefficient change values ​​can also be represented in the frequency domain by using a Fast Fourier Transform.

Line 251, fig 7. You can also represent the coefficient change values ​​in the frequency domain using a Fast Fourier Transform.

Line 259, Figure 8: Coordinate changes can be presented in the frequency domain and describe how harmonics change in the time domain and the frequency domain.

Line 273, figure 9. Same as above. The effectiveness of the method will be more visible in the case of the presentation of results in the frequency domain of changes in the position of satellites in orbit.

Regards.

Round 2

Reviewer 1 Report

It is a good work for this manuscript.

Reviewer 3 Report

Well done. Best regards.